# Study on Dissolution and Modification of Cotton Fiber in Different Growth Stages

**DOI:** 10.3390/ma15072685

**Published:** 2022-04-06

**Authors:** Xiaonan Deng, Sihong Ye, Lingzhong Wan, Juan Wu, Hui Sun, Ying Ni, Fangzhi Liu

**Affiliations:** Cotton Research Institute of Anhui Academy of Agricultural Sciences, Hefei 230001, China; xn_deng@foxmail.com (X.D.); bx_8969@163.com (L.W.); wujuan@aaas.org.cn (J.W.); sunhui@foxmail.com (H.S.); ny@163.com (Y.N.); ahaqliu@163.com (F.L.)

**Keywords:** cotton, cellulose, deep eutectic solvent, fiber development

## Abstract

Cotton fibers with ultra-high purity cellulose are ideal raw materials for producing nanocellulose. However, the strong hydrogen bond and high crystallinity of cotton fibers affect the dissociation of cotton fibers to prepare nanocellulose. The structures of two kinds of cotton fibers (CM and XM) in different growth stages from 10 to 50 days post-anthesis (dpa) were studied by Fourier transform infrared spectroscopy (FTIR) and X-ray diffraction (XRD). In the process of cotton fiber growth, the deposition rate of cellulose macromolecules firstly increased and then stabilized. Then, the surface morphology, the chemical composition, and the crystal structure of the nanocellulose prepared from cotton fibers with different growth stages by deep eutectic solvent, a green solvent, were characterized by Transmission Electron Microscope (TEM), scanning electron microscopy (SEM) analysis, XRD, and Thermo Gravimetry (TG). The growth time of cotton fibers affected the properties of prepared nanocellulose, and nanocellulose obtained from cotton fibers at about 30 dpa had less energy consumption, higher yield, and milder reaction conditions.

## 1. Introduction

Cotton is an important economic crop in China, and cotton fiber is one of the natural resources with the highest fiber purity. Cotton fiber is a single-celled fiber formed by the development of ovules epidermis cells [1,2]. After initial differentiation, it rapidly elongates and generally no longer divides [3]. The final fiber length of fiber cells can reach thousands of times its diameter. The formation and development process of cotton fiber can be divided into four stages; this involves fibrous primal cell differentiation and processes (initiation), primary wall elongation, secondary wall thickening, and maturation overlap. About one-third of the epidermal cells of the cotton ovule can develop into cotton fibers, which mature after about 50 days and eventually become dehydrated and twisted bands of 20~60 mm in length. Cotton fiber is composed of more than 95% cellulose [4]. Cotton fiber has excellent physical and mechanical properties [5], high strength, good toughness, high yield, and a short growth cycle. It is an ideal renewable plant resource. Studies have proven that the different structure of cotton cellulose leads to the great difference in its physical, mechanical, and chemical properties. The crystal structure of cellulose in cotton fiber during fiber development from 21 to 60 dpa was studied [6,7]. In addition, the results show that crystallinity and grain size significantly increase between 21 and 34 dpa, which is the first half of the secondary cell wall thickening process. However, the existence of strong hydrogen bonds between cellulose chains limits the easy use of cotton fibers in nanocellulose production, and there are few articles studying the growth process of cotton fiber in different stages. Therefore, a systematic study of the structure of the growth process of cotton fiber, the dissolving and efficient methods of cotton fiber in textile engineering, pulp and paper, and biomass fuel conversion, such as cotton fiber-based nanomaterials preparation processing, has very profound significance.

In 2003, the research group of Abbott [8] first proposed a low-melting solvent composed of urea and choline chloride and named it as a deep eutectic solvent. Deep eutectic solvents and ionic liquids have similar physical and chemical properties, therefore they are called “ion-like liquids” or “eutectic ionic liquids” [9]. Deep Eutectic Solvents (DESs), formed by mechanical mixing of a hydrogen bond donor (HBD) and a hydrogen bond acceptor (HBA) in a certain molar ratio, are generally transparent liquid mixtures at lower temperatures. Hydrogen bond receptors are generally quaternary ammonium salts, which form DESs [5,10,11] by shielding charges near some HBDS. Due to the hydrogen bonding and van der Waals forces between components, the recrystallization ability of raw materials decreases, thus reducing the melting point of the system [12]. The strength of the hydrogen bond is related to phase transition temperature, stability, and solvent properties. Recently, Liu [13] reported that microwave-assisted DES treatment can effectively separate lignin and hemicellulose from wood and quickly obtain high-purity cellulose, which provides a good prospect for CNC production. DES is also used for de-crystallization and modification of cellulose [14,15,16,17,18].

In this paper, the cellulose deposition and organization during different phases of cotton fiber development from 10 dpa to 40 dpa are studied by Fourier transform infrared (FTIR), X-ray diffraction (XRD), and SEM spectroscopy. Then, the cotton fibers at different growth stages were dissolved in deep eutectic solvent, and nanocellulose from cotton fibers at different growth stages was prepared by ultrasound. The structural properties of nanocellulose were analyzed by TG, XRD, SEM, and TEM. This study will provide theoretical support for the high-value utilization of cotton fiber.

## 2. Materials and Methods

### 2.1. Materials

Oxalic acid dihydrate (purity: >99%) and choline chloride (purity: >98%) were obtained from Sinopharm Chemical Reagent Co., Ltd. (Shanghai, China). Sodium chloride (purity: >99%) was also obtained from Sinopharm Chemical Reagent Co., Ltd. (Shanghai, China). Two kinds of cotton, CM and XM, were applied in the experiment (CM and XM, Anhui Academy of Agricultural Sciences, Anqing, Anhui, China (30°31′ N, 117°06′ E)), as shown in Figure 1. The average day and night from May to October was 22.4 °C, and the total rainfall in 2021 was 1686 mm.

### 2.2. Methods

#### 2.2.1. Cotton Fiber Sampling Method

Two kinds of cotton were used as materials, and single flowers were marked on flowering day (0 dPa) and 10 growth bolls were harvested at different growth stages. These stages represent the major developmental stages of cotton fibers. The pericardium was immediately removed, and the isolated ovules were transferred to a cryogenic flask and stored in a cryogenic biological storage system filled with liquid nitrogen until analysis.

#### 2.2.2. DES Treatment

DES with a molar ratio of 1:2 was prepared with 63.0 g oxalate dihydrate and 69.8 g choline chloride, and stirred continuously for 60 min in a glass reactor at 95 °C. About 0.1 g of treated cotton fiber was added to the reactor. After the reaction, 200 mL of deionized water was added to the reactor to reduce the viscosity of the mixture. The treated cotton cellulose was cleaned and filtered. The pretreated fibers were first mixed with water at a concentration of 0.5 wt.%. The suspension was then treated with a JY99-IIDN ultrasonic generator (Ningbo Xinzi Biotechnology Co., Ltd., Ningbo, China) equipped with a 25 mm cylindrical titanium alloy probe. The ultrasonic frequency was 20 kHz, the output power was 1000 W, and the ultrasonic time was 30 min.

### 2.3. Characterization and Measurements

The FTIR spectra were recorded using a Nicolet Magna 560 FTIR instrument (Thermo Fisher Scientific Inc., Waltham, MA, USA) with a diamond attenuated total reflectance.

X-ray diffraction analysis was performed on a Bruker D8 ADVANCE instrument with copper target radiation. The acceleration voltage was 50 kV, the scanning range 2θ was from 10° to 50°, the scanning speed was 5°/min, and the tube current was 40 mA. Crystallinity was calculated according to the following formula.
(1)$$\mathrm{CrI}=\left[\frac{I_{200}-I_{\mathrm{am}}}{I_{200}}\right]\times 100$$

In the formula, I_200_ is the peak intensity of the diffraction of the main crystal plane at 22.7°; I_am_ is the diffraction intensity of the amorphous region at Angle 2θ close to 18°.

The morphology of the pretreated cotton fibers in different growth stages was observed using a JSM-7500F scanning electron microscope (SEM) (JEOL, Tokyo, Japan) at 15 kV.

TEM observation on the CNFs was performed using Thermo Fisher FEI Quanta FEG 450 (Thermo Fisher Scientific Inc., Waltham, MA, USA) at 120 KV acceleration voltage. A drop (5 µL) of diluted CNFs slurry was dropped on a carbon-coated electron microscope grid, and then negatively stained with 1 wt.% phosphotungstic acid solution to enhance the contrast of the image. The size of the nanofibrilated cellulose was measured by transmission electron microscope.

## 3. Results and Discussion

### 3.1. FTIR Analysis

FTIR analysis was performed on fibers from CM and XM. Figure 2 shows typical FTIR spectra acquired at different developmental stages. It indicates that in the early stage of cotton fiber development (from 10 dpa to 20 dpa), most of the cellulose is immature, the fiber ovules contain a lot of water, and the characteristic peak of cotton fiber is not obvious. As the tissue of the cotton cellulose develops (the crystallinity increases from 25% to 75%), the adsorption of water decreases as water molecules have difficulty accessing the OH group, resulting in a typical cotton fiber profile at maturity. The characteristic absorbance peaks of the cellulose were 1372, 1165, 1117, 1062, and 897 cm^−1^, respectively, and the absorption peaks were consistent with the structure of cotton cellulose (Figure 2).

### 3.2. XRD Analysis

Figure 3 showed the diffraction patterns of fibers harvested from CM and XM cultivar at different stages of development. At early stages of fiber development (CM-1 and CM-2), the typical X-ray diffraction pattern of cellulose is not clearly evident. This is due to the immature development of cotton cellulose crystals at the early stage of fiber growth. At CM-3 dpa, the relative intensity of the 200 lattice starts increasing and the 1–10 and 110 can be seen in the spectrogram. Figure 4 shows the evolution of the crystallite size as a function of 45 dpa for fibers harvested from CM and XM cultivars. The crystalline size of CM fibers was about 15% before 10 dpa and increased to 46% at 20 dpa. This rapid growth occurred in less than 10 days. Fibers harvested from the XM variety had a crystallinity of 14% before 10 dpa. After about 30 dpa, the fiber crystallinity of the two varieties was consistent. The crystallinity did not change after 40 dpa.

### 3.3. Nanofibrillate Cellulose Prepared from Cotton Fibers at Different Growth Stages

The cotton fibers at different growth stages were used as raw materials to prepare nanocellulose, which was pretreated by deep eutectic solvent composed of choline chloride/oxalic acid. The treated cotton fibers were pulverized by ultrasonic wave to obtain micro-nano cotton cellulose.

The yield and the component analysis of cotton fiber pretreated with deep eutectic solvent are shown in Table 1. It can be seen that at a temperature of 90°, the yield significantly decreases with the increase in cotton maturity, and the treatment conditions become high. In the third stage of cotton fiber maturation, the yield of pretreated cotton fiber can reach 77%. The component content analysis results of fibers shows that the content of cellulose in different stages of cotton fiber is about 60–90%, and the cotton fiber content is higher when the maturity is higher. As seen from Figure 5, the color of pretreated cotton fiber deepened with the treatment time increasing, indicating that cotton cellulose degradation was intensified, and its color change was obvious under long-term eutectic solvent treatment. Long-term deep eutectic solvent treatment had a significant effect on cotton fiber depolymerization.

### 3.4. XRD Analysis of Different Stages of Nanocellulose Treated with Deep Eutectic Solvent

The X-ray diffraction (XRD) method was used to study the crystal structure and relative crystallinity of nanocellulose. As shown in Figure 6, the characteristic diffraction peaks of nanocellulose at 2θ of 14.2°, 16.8°, and 22.4° at different growth stages indicate that their crystal structure is cellulose I type. The results showed that the deep eutectic solvent combined with ultrasonic treatment did not change the crystal structure of cellulose. As an important index affecting the thermal stability of cellulose, the crystallinity was calculated by Segal method (Segal et al. 1959), and the crystallinity of cotton fiber was 60%. The crystallinity of cotton fibers increased by 5% to 15% compared to the raw material at different stages, which was obviously due to the dissolution of the primary wall or the removal of oligosaccharides.

### 3.5. TG Analysis of Different Stages of Nanocellulose Treated with Deep Eutectic Solvent

The TG curves of nanocellulose obtained from different stages of cotton fibers can be seen in Figure 7. This can not only reflect their thermal stability, but also reflect the difference of crystallinity and molecular weight between them. All samples showed similar TG curves, and the initial degradation temperatures (T_onset_) of cotton cellulose at different growth stages were slightly different. The initial degradation temperature of the original cotton fiber was approximately 300 °C, while the initial degradation temperature of the prepared nanofibrillated cellulose was raised to approximately 305 °C. Compared with original cotton cellulose, nano-granulated cellulose has a smaller size, larger specific surface area, and lower molecular weight, thus its thermal stability is greatly affected

### 3.6. SEM Analysis of Different Stages of Nanocellulose Treated with Deep Eutectic Solvent

The SEM of nanocellulose obtained from different stages of cotton fibers can be seen in Figure 8. The SEM shows that deep eutectic solvent after treatment at different stages of cotton fiber’s surface microstructure was similar to the fiber development of cotton fiber in prophase that was incomplete. At the same time, it was found that a large number of fibrous structures appeared on the surface of cellulose in the later growth period. The results showed that under the condition of certain depolymerization of cotton fiber, part of the cellulose chain was broken, and then NFC was prepared.

### 3.7. TEM Analysis of Different Stages of Nanocellulose Treated with Deep Eutectic Solvent

The size of NFC at different growth stages prepared by deep eutectic solvent pretreatment was analyzed by TEM as shown in Figure 9. The results showed that the cotton fibers at different growth stages could be prepared by ultrasonic nanocellulose. In the third stage of cotton fiber development, the treatment effect was obvious, the diameter of the nanocellulose was small, and the pretreated fiber could be completely transformed into cellulose nanocrystalline. With the growth cycle of cotton fibers increasing, the cotton fibers dissociated the filaments obviously under treatment with deep eutectic solvent, which leads to the system color deepening and the decreases in size of the nanocellulose. Comprehensively considering the yield and energy consumption in the preparation process of the nanocellulose, 30 dap cotton fiber is suitable for preparation of nanofilament cellulose.

## 4. Conclusions

### 4.1. Structural Changes of Cotton Cellulose at Different Growth Stages

In order to better clarify the accumulation and structure change of cellulose during cotton growth, FTIR and XRD were used to analyze the structure and crystallinity of cotton fiber. Both CM and XM cotton fibers showed characteristic absorbance peaks of the cellulose from the analysis of FTIR. The major change in cellulose organization occurs between 16 and 30 dpa. This phase represents the active cellulose deposition phase during the secondary cell wall synthesis. The crystalline size of cotton fibers increased fast in less than 10 dpa, while after 30 dpa, the cotton fibers of CM and XM had a consistent crystalline size.

### 4.2. Preparation of Nano-Cotton Cellulose from Cotton at Different Growth Stages

The properties of the nanocellulose prepared from cotton at different growth stages showed that the relatively optimal treatment condition is the reaction temperature of cotton fibers at 90 °C in eutectic solvent. Considering the yield and the process energy consumption, cotton fibers at about 30 days after flowering were suitable for the preparation of nanocellulose. The maximum yield of nanocellulose can be as high as 74.2%.

## Figures and Tables

**Figure 1 materials-15-02685-f001:**
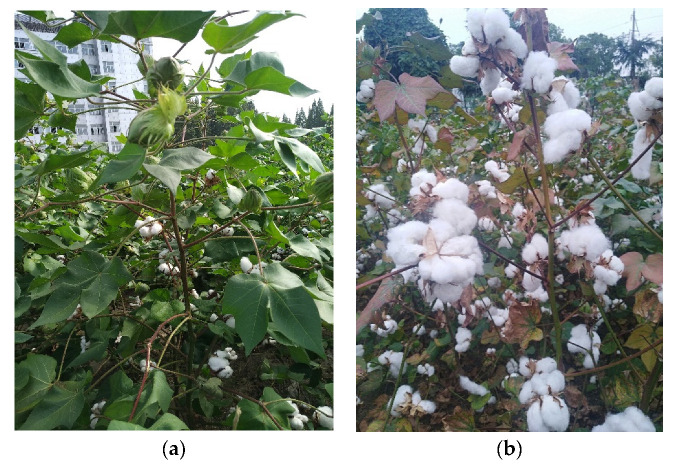
The two varieties cotton. (**a**): CM, (**b**): XM.

**Figure 2 materials-15-02685-f002:**
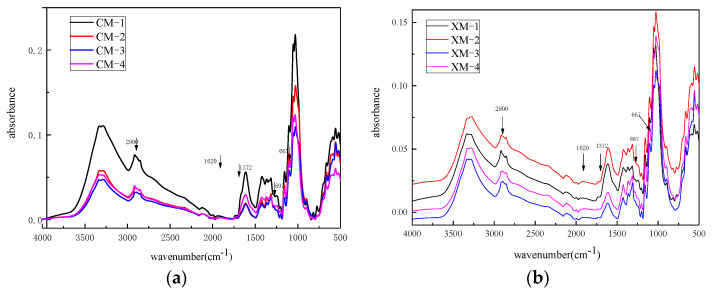
FTIR spectra of fibers harvested at different developmental stages from CM (**a**) and XM (**b**) cultivar.

**Figure 3 materials-15-02685-f003:**
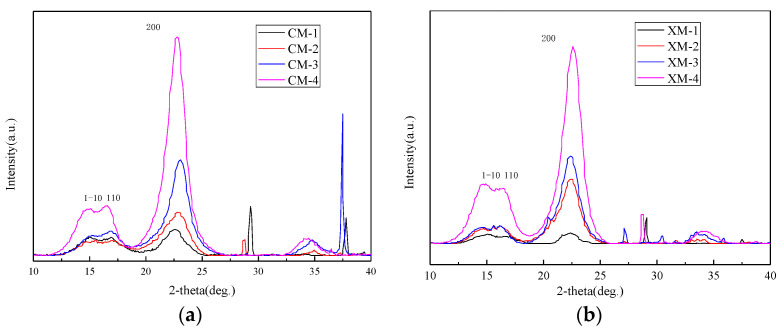
X-ray diffraction patterns of fibers harvested from (**a**) CM and (**b**) XM.

**Figure 4 materials-15-02685-f004:**
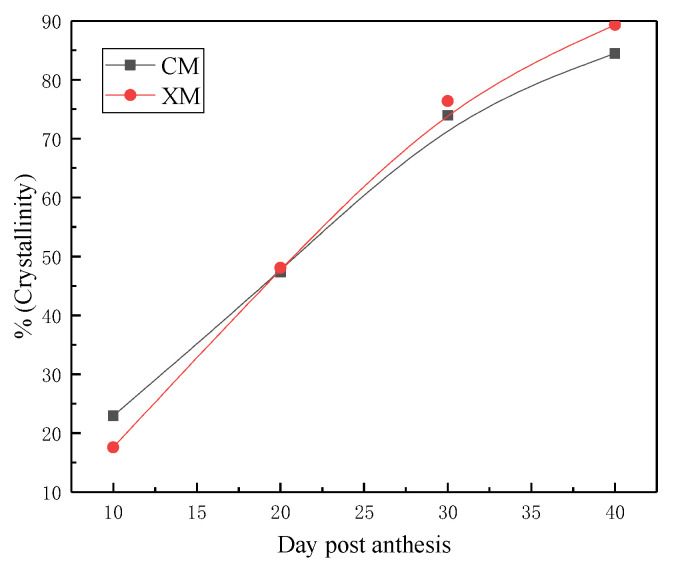
Crystallinity percentage of fibers harvested from CM and XM at different growth stages.

**Figure 5 materials-15-02685-f005:**
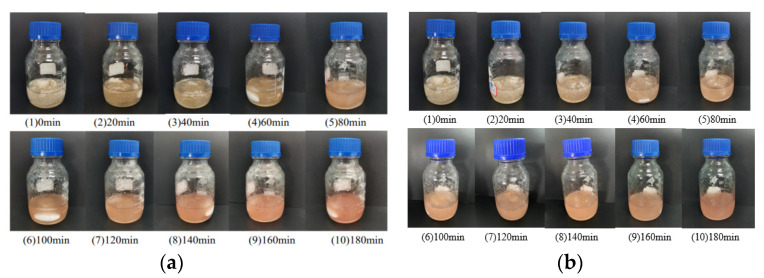
The dissolution process of (**a**): XM-3, (**b**) CM-3 in deep eutectic solvent.

**Figure 6 materials-15-02685-f006:**
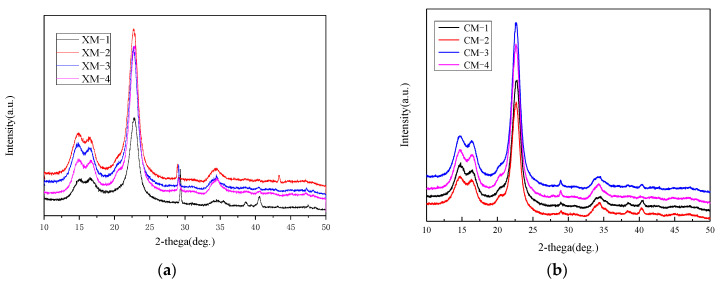
X-ray diffraction patterns of fibers harvested from (**a**) XM and (**b**) CM.

**Figure 7 materials-15-02685-f007:**
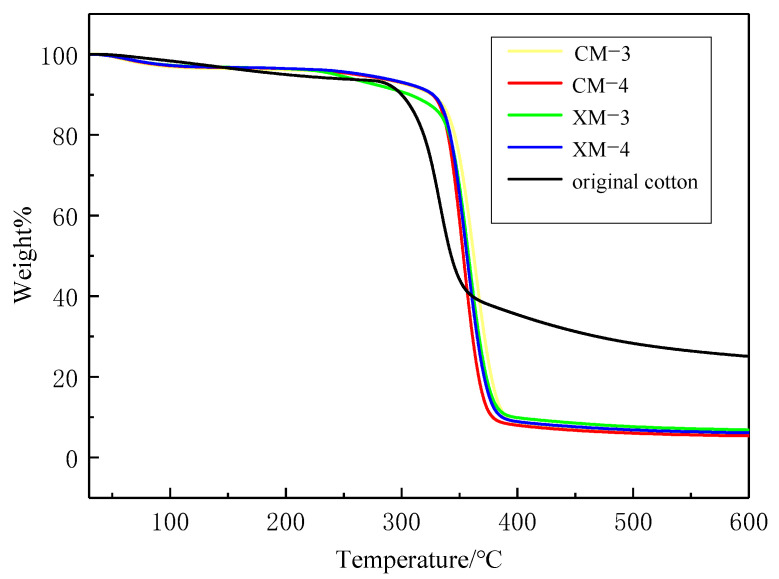
The TG curves of the original cotton fibers and the modified cotton fibers at different developmental stages from original cotton, CM-3, CM-4 and XM-3, XM-4 cultivar.

**Figure 8 materials-15-02685-f008:**
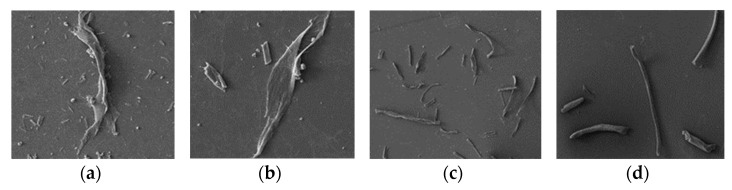
Morphology of the pretreated cotton fiber at different growth stages, (**a**): XM-1, (**b**): XM-2, (**c**): XM-3, (**d**): XM-4.

**Figure 9 materials-15-02685-f009:**
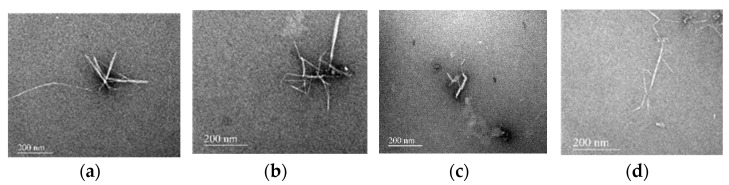
TEM images of the pretreated cotton fiber at different growth stages, (**a**): XM-1, (**b**): XM-2, (**c**): XM-3, (**d**): XM-4.

**Table 1 materials-15-02685-t001:** Yield analysis of cotton fiber pretreated with eutectic solvent in different growth stages.

Cotton Varieties	Temperature/°C	Dissolution Time/min	Power/W	Yield/%
XM 1	90	270	1200	60.40
XM 2	90	270	1200	66.73
XM 3	90	220	1200	76.93
XM 4	90	180	1200	84.06
CM 1	90	270	1200	53.85
CM 2	90	270	1200	62.50
CM 3	90	220	1200	74.42
CM 4	90	180	1200	82.72

## Data Availability

Not applicable.

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
