# Peer review of "Study on Dissolution and Modification of Cotton Fiber in Different Growth Stages"

_materials, 2022, doi:10.3390/ma15072685_

Round 1

Reviewer 1 Report

Comments are attached.

Reviewer 2 Report

The submitted article presents a study on dissolution and modification of cotton fiber in different growth stages. The structure of cellulose in different cotton growth stages of two varieties was studied by FTIR and XRD analysis. Then the surface morphology, the chemical composition, and the crystal structure of nanocellulose prepared from cotton fibers were characterized by TEM, SEM analysis, XRD, and TG to prepare different growth stages cotton fiber nanocellulose by green solvent deep-eutectic solvent. The authors concluded that the cotton fiber about 30 dpa of nanocellulose has less energy consumption, higher yield and mild reaction conditions.

Specific points requiring attention are detailed below.

Comment 1. The introduction of the manuscript includes only 15 references. The known in the literature information is not sufficiently and profoundly presented.

Comment 2. The English language needs considerable improvement.

Comment 3. The use of the Fourier transform infrared spectroscopy, X-ray diffraction analysis and SEM and TEM observation for cellulose study is a routine study. The use of deep eutectic solvent for dissolution of cellulose or cotton is already reported in literature. The overall novelty the manuscript is missing.

Comment 4. The designation of CM-1, CM-2, CM-3, CM-4, XM-1, XM-2, XM-3 and XM-4 in the Figures 1 and 2 is missing.

Comment 5. The differences in the dissolution process presented in Figure 5 are hard to be seen.

Comment 6. In Figure 8 the magnification of the SEM image is missing. There is no scale bar.

Comment 7. In the whole article including introduction and results section the critical discussion is missing.

Comment 8. Higher magnifications in SEM analysis should be used to see the cotton fiber structure.

Comment 9. Higher magnifications in TEM analysis should be used to see the cotton fiber structure.

Reviewer 3 Report

This research investigated the nanocellulose production from cotton fibers at different growth stages. While this paper can be of interest to many readers of this field, the academic presentation of this paper must be extensively improved for publication. 

  1. Abstract: extensive language editing is required. it was very hard to understand the abstract
  2. The Results and Discussions part overall needs to be more specific. For instance, in Figure 2, the authors indicated certain FTIR peaks. However, there was almost no explanation about how each peak is associated with the cotton cellulose chemistry and how FTIR spectra differ per the growth stages. 
  3. Figure 8 is missing scale bars. To me, all the images in Figure 8 and Figure 9 seem similar. Perhaps it's because of the missing scale bars, but the authors need to clarify how they concluded that they saw significant differences in nanocellulose structures in different growth stages. 
  4. An extensive level of English editing is required since many parts of the manuscript are unclear

Round 2

Reviewer 2 Report

Nevertheless, the authors have improved some of the specific points like the inclusion of three more references and the addition of bars to the SEM micrographs, the overall novelty and originality in the revised manuscript are not clearly stated. The manuscript needs more English editing and better writing. For instance: the Introduction and Conclusion section contained the same sentence: The cotton fiber about 30 dpa of nanocellulose has less energy consumption, higher yield and mild reaction conditions. Critical discussion of the obtained results is missing as well. Therefore, I recommend that the manuscript is not accepted for publication in Materials.

Author Response

First of all, we would like to show our respect and appreciation to the reviewers since we could never improve our research without their great help. After receiving the reviewer’s comments, we read the comments carefully and revise the manuscript as follows.

Reviewer 3 Report

The authors addressed most of my comments. However still language editing is required.

Author Response

First of all, we would like to show our respect and appreciation to the reviewers since we could never improve our research without their great help. After receiving the reviewer’s comments, we read the comments carefully and revise the manuscript.
